# 2-Azidobenzaldehyde-Enabled Construction of Quinazoline Derivatives: A Review

**DOI:** 10.3390/ijms26188955

**Published:** 2025-09-14

**Authors:** Weiqi Qiu, Desheng Zhan, Xiaoming Ma, Xiaofeng Zhang

**Affiliations:** 1Department of Chemistry, Boston College, 2609 Beacon Street, Chestnut Hill, MA 20467, USA; 2College of Chemistry, Changchun Normal University, Changchun 130032, China; 3School of Pharmacy, Changzhou University, Changzhou 213164, China; 4Department of Cancer Biology, Dana-Farber Cancer Institute, Harvard Medical School, Harvard University, Boston, MA 02215, USA

**Keywords:** 2-azidobenzaldehyde, azomethine imines, 1,3-dipolar cycloaddition, quinazolines, heterocyclic, multicomponent reactions, diversity-oriented synthesis

## Abstract

Quinazoline is a privileged heterocyclic scaffold commonly found in numerous pharmaceuticals and bioactive natural products, known for its diverse biological activities. The pursuit of efficient and versatile synthetic methods to produce quinazoline derivatives remains a central focus for organic and medicinal chemists, owing to the therapeutic potential of these compounds. This paper reviews the innovative use of 2-azidobenzaldehyde-enabled annulation strategies for the synthesis of quinazoline derivatives, including quinazolin-4(3H)-one, 2,3-dihydroquinazolin-4(1H)-one, 3,4-dihydroquinazoline, 3,4-dihydroquinazoline-2(1H)-thione, and 1,2,3,4-tetrahydroquinazoline. Emphasizing both the mechanistic insights and practical advantages, this review highlights the efficacy and applicability of these methods in the domain of heterocyclic chemistry, providing an invaluable framework for future drug discovery and development efforts.

## 1. Introduction

Quinazoline derivatives are an important class of nitrogen-containing heterocycles that have garnered considerable attention in drug discovery and medicinal chemistry [1,2,3]. Their unique chemical structure allows for versatile modifications, leading to a wide array of pharmacological activities [4,5]. The quinazoline scaffold serves as a core structure in numerous therapeutic agents due to its ability to interact with diverse biological targets. Quinazoline derivatives have been pivotal in drug discovery, serving as key components in the development of treatments for various diseases, such as quinazoline-based drugs in the market (Figure 1) [6,7,8]. Their ability to inhibit critical enzymes and receptors has made them attractive candidates for anticancer, antiviral, antibacterial, and anti-inflammatory drugs [9,10,11]. For instance, several quinazoline-based compounds have been developed as epidermal growth factor receptor (EGFR) inhibitors for treating cancers, particularly non-small lung cancer cells, by impeding tumor growth and proliferation [12,13,14,15].

The synthesis of quinazoline derivatives has seen significant advancements with numerous methodologies developed to efficiently construct these biologically important heterocyclic scaffolds [1,16,17,18,19]. A prominent trend in the synthesis of quinazoline derivatives is the use of transition-metal-catalyzed reactions, which have emerged as indispensable tools in organic synthesis due to their ability to streamline complex procedures, increase yields, and reduce reaction times [20]. Advances in synthetic techniques, including microwave-assisted synthesis [21], green chemistry approaches [22,23], and multicomponent reactions [24,25,26,27], have made it possible to produce quinazolines with high yields and purity.

Quinazoline derivatives, a fascinating class of compounds known for their diverse biological activities, are generally categorized into several key types: 3,4-dihydroquinazoline [28,29], 2,3-dihydroquinazolin-4(1H)-one [30,31,32], 4-quinazolinone [33,34,35,36], 3,4-dihydroquinazoline-2(1H)-thione [37,38], and 1,2,3,4-tetrahydroquinazoline [39,40], which are seen in Figure 2. The synthetic innovations associated with these categories have significantly broadened their applicability, allowing for the creation of novel compounds specifically designed to meet various therapeutic needs.

Among these, 2-azidobenzaldehyde stands out as a particularly versatile synthetic intermediate. This material plays a critical role in the construction of various heterocyclic systems, such as quinolines [41], and quinazolines, owing to its reactive azido group and aldehyde functionalities. The exploration of 2-azidobenzaldehyde-enabled synthesis techniques opens new avenues for the development of quinazoline derivatives, as detailed in this article. Through these innovative methodologies, researchers continue to expand the potential applications of quinazoline derivatives in medicinal chemistry and beyond, addressing complex therapeutic challenges with tailored molecular solutions.

## 2. Synthesis of Quinazoline Derivatives

### 2.1. Quinazolines

Quinazoline serves as a crucial pharmacophore in drug discovery. Numerous substituted quinazoline derivatives exhibit remarkable bioactivities, with some having received approval from the Food and Drug Administration (FDA) for clinical use (Figure 1) [42]. An example of synthesizing quinazoline 3-oxides **4** can be found in the Pd(II)-catalyzed three-component reaction (3-CR) involving 2-azidobenzaldehyde **1**, isocyanide **2**, and hydroxylamine hydrochloride **3** in a one-pot procedure reported by the Sawant group (Figure 1) [43]. This approach offers significant advantages over traditional methods, which typically rely on prefabricated substrates generated through multistep syntheses. Conventional techniques often suffer from drawbacks such as low yields, harsh reaction conditions, limited substrate scope, and the use of expensive starting materials [44,45,46,47]. In contrast, the Pd(II)-catalyzed strategy provides a more efficient and streamlined pathway to quinazoline 3-oxides **4** with 15 examples in the range of 71–91%.

In general, isocyanides are highly versatile chemicals that enable the rapid assembly of complex molecules. However, their characteristic odor and toxicity, as well as their tendency to undergo undesired side reactions under harsh conditions, require careful handling and consideration of such environmental impact, regioselectivity, and scalability in synthetic applications [48]. Thus, for the validation of mechanistic pathways, authors confirmed that 3-CR proceeds predominantly through the generation of compounds **5** and **6,** involving the azide–isocyanide denitrogenative coupling/condensation/6-exo-dig cyclization by a series of control experiments, and denied a pathway via compound **7a** (Figure 2). Pd(II)-catalyzed mechanism was proposed in Figure 3; initially, coordination of 2-azidobenzaldehyde **1a** and isocyanide **2a** with Pd(OAc)_2_ generates intermediate **8** in mild conditions. This intermediate undergoes nitrogen extrusion to form nitrene intermediate **9**. Subsequently, intramolecular isocyanide transfer over the nitrene occurs in a concerted manner, yielding carbodiimide **5a**. This reactive carbodiimide then enters the second catalytic cycle, coordinating with palladium metal. During this cycle, the aldehydic functional group of **5a** condenses with hydroxylamine **3**, resulting in the loss of a water molecule that is trapped by 4 Å molecular sieves. This condensation furnishes hydrazone **12**, which subsequently undergoes 6-exo-dig cyclization to produce quinazoline-3-oxide **4a**.

Quinazoline-derived azomethine imines (QAIs) have emerged as a compelling class of compounds in this field due to their high reactivity. These QAIs, based on the quinazoline scaffold, can effectively serve as 1,3-dipoles in cycloaddition or formal cycloaddition reactions [49]. This enables the construction of diverse quinazoline-fused polycyclic compounds, highlighting their versatility and potential in synthetic chemistry. Compared to traditional methods that rely on a three-step process involving 2-nitrobenzaldehyde for the preparation of functional azomethine imines **15** [50,51,52]. The Sawant group has developed a more practical approach. They reported a Pd-catalyzed three-component reaction (3-CR) protocol using 2-azidobenzaldehyde **1**, tert-butyl isocyanide **2**, and sulfonyl hydrazide **14** in tetrahydrofuran to synthesize tert-butylamino-substituted azomethine imines **15** in 65–86% yield (Figure 4) [53]. This 3-CR method via cross-coupling/condensation/condensation/6-exo-dig cyclization is more efficient in terms of operational simplicity and effectiveness, despite the presence of a substituent on the pyrimidine ring.

### 2.2. 3,4-Dihydroquinazolines

3,4-Dihydroquinazoline-based compounds have garnered significant attention in both natural product chemistry and pharmaceutical research due to their diverse biological activities and therapeutic potential, such Anagrelide, Letermovir, Quazinone, Vasicine, Linagliptin and Deoxyvasicine (Figure 3). The presence of the 3,4-dihydroquinazoline core in natural products and marketed drugs often contributes to their unique mechanisms of action and biological efficacy, making them valuable leads in drug discovery [28,29,54,55].

In contrast to the reported complex and intricate synthetic protocols [56,57,58,59,60], the synthesis of 3,4-dihydroquinazoline derivatives via Ugi-initiated approaches [61,62,63,64], and azomethine imines-promoted one-pot processes [49] is systematic, straightforward, and easy to implement. These methods are associated with detailed and specific mechanistic studies, ensuring good reproducibility and offering potential for further in-depth exploration. By enhancing operational efficiency, these methodologies provide a robust platform for advancing research, leading to deeper levels of understanding and innovation in the field.

Among these efforts, the Sawant group reported a series of innovative syntheses of 3,4-dihydroquinazoline derivatives, enabled by 2-azidobenzaldehyde and executed through a four-component reaction (4-CR). This process involves the generation of azomethine imines **15**, followed by a 1,3-dipolar cycloaddition to drive the diversity-oriented synthesis (DOS) [65,66] of 3,4-dihydroquinazoline derivatives, providing the means to explore uncharted chemical and biological spaces, ultimately driving forward the discovery of new and effective therapeutics. Initially, they demonstrated a 4-CR using four versatile privileged synthons: 2-azidobenzaldehyde **1**, isocyanide **2**, sulfonyl hydrazide **14a**, and alkynes **17**. This reaction, promoted by the transition metal catalysts Pd(OAc)_2_ and AgOTf, yielded pyrazolo[1,5-c]quinazolines **18** (Figure 5) [67]. The 4-CR process efficiently generates five new chemical bonds, producing diverse compounds **18**, with 32 examples yielding between 46 and 97% in a single operation. Moreover, substituting alkynes **17** replaced by electron-deficient alkenes **19**, such as acrylates and acrylonitrile, facilitates the one-step synthesis of tetrahydropyrazolo[1,5-c]quinazolines **20**. This 4-CR mechanism was validated through a stepwise synthetic process, demonstrating that azomethine imines **15** was formed via the azide–isocyanide denitrogenative coupling, condensation, and 6-exo-dig cyclization, then followed by 1,3-dipolar cycloaddition to yield compounds **18** and **20**. These powerful molecules underwent cell viability assays, revealing excellent cytotoxic effects and strong inhibition of EGFR, with docking studies highlighting hydrogen bonding interactions with key amino acid residues, namely Met769, Glu738, and Thr766 [67].

Subsequently, the Sawant group investigated the scalability of 4-CR using 1,3-dipolar cycloadditions of azomethine imines **15**, employing various dipolarophiles in a one-pot approach. Authors synthesized azomethine imines **15a** from a 3-CR of 2-azidobenzaldehydes **1**, isocyanides **2**, and tosyl hydrazides **14a**. These were then used to produce pyrazolo[1,5-c]quinazolines **22** and **24** by incorporating dipolarophiles like **21** and **23** in the 1,3-dipolar cycloaddition process (Figure 6) [68]. Furthermore, one-pot two-step reaction conditions were optimized using 1,4-diazabicyclo[2.2.2]octane (DABCO) and DABCO/I_2_ under palladium catalysis at 100 °C for 2 h, yielding compounds **22** with 13 examples ranging from 73 to 93%, and compounds **24** with 5 examples, yielding between 65 and 87%.

Additionally, a Pd-catalyzed 4-CR involving 2-azidobenzaldehyde **1**, isocyanides **2**, sulfonyl hydrazides **14**, and 2-(trimethylsilyl)-phenyltriflates **25**, the latter serving as an aryne precursor, was explored for the cascade synthesis of fluorescent indazolo[2,3-c]quinazolines **26** (Figure 7) [69]. This resulted in 16 examples with yields between 63 and 82%. These compounds demonstrated absorption in the visible region, high quantum yield fluorescence, and excellent photostability. This cascade 4-CR encompasses three sequential transformations: (1) palladium-catalyzed formation of azomethine imine; (2) cyclocondensation with hydrazides; and (3) carboamination of aryne [70]. Azomethine imines **15** were synthesized by 3-CR in hand; the Sawant group further explored diverse one-pot 4-componet synthesis by offering nitroolefins **27** and allenoates **29** as dipolarophiles to make 1-nitro-2-aryl-1,2,3,10b-tetrahydropyrazolo[1,5-c]quinazolines **28** and 2-methylpyrazolo[1,5-c]quinazolines **30** in medium yields, respectively (Figure 8) [53]. This 4-CR was also offered in the DOS of compounds **32** and **34** through using *α*-halo hydroxamates **31** and cyclic ketones **33** reacting with versatile azomethine imines **15** (Figure 8).

DOS is a strategic approach in organic synthesis designed to explore novel reaction pathways. It plays a crucial role in drug discovery by generating structurally diverse compounds, thereby identifying potential molecules with a wide range of biological activities [53,65,66]. Zhang group exemplified this approach by using 2-azidobenzaldehyde **1** to promote 1,3-dipolar cycloaddition with amino esters **35** and maleimides **36**, resulting in versatile pyrrolidine adducts **37**. These adducts contain dual functional NH and N_3_ groups, enabling a range of DOS applications through various reaction pathways. Notable pathways include click chemistry, radical reactions, and Staudinger/aza-Wittig reactions, which effectively connect the NH and N_3_ groups and expand the diversity of the resulting compounds [71,72]. Recently, Ma and colleagues reported a cascade reaction process through Pd-catalyzed azide–isocyanide coupling/cyclization/lactamization reactions using dual functional intermediates **37** for the synthesis of tricyclic guanidine-containing polyheterocycles **38** with 30 examples in the scale of 47–82% yields (Figure 9) [73].

In 2010, Ding group reported a stepwise Biginelli/Staudinger/aza-Wittig process to construct 3,4-dihydroquinazoline derivatives, which involves the preparation of dual functional adducts **41** by 3-CR with 2-azidobenzaldehyde **1a**, ethyl acetoacetate **39**, and urea **40**, followed by one-pot Staudinger and aza-Wittig reactions to give carbodiimides **43** without the isolation of iminophosphoranes **42**, then cyclized easily to afford pyrimido[1,6-c]quinazolin-4-ones **44** in moderate to good overall yields in the presence of catalytic amount of potassium carbonate in acetonitrile (CH_3_CN) at room temperature (Figure 10) [74]. This strategy was also applied to make compounds **46** with 17 examples in the scale of 61–92% yields by the reaction of iminophosphoranes **42** with acyl chloride.

Ding group reported a second example of 3,4-dihydroquinazoline synthesis using an Ugi/Staudinger/aza-Wittig sequence. This method produced 33 examples of compound **51** with yields ranging from 35% to 93%. Initially, adducts **49** were synthesized with yields of 66% to 92% through a 4-CR (Ugi) involving substrates **1a**, **2**, **47**, and **48**. This was followed by a one-pot Staudinger/aza-Wittig process, as depicted in Figure 11 [64]. Some of the products **51** with dual functional sites prepared by Ugi reaction of 2-bromobenzenamine **48a**, cinnamic acids **47**, 2-azidobenzaldehyde **1a,** and isocyanide **2**, could be implemented in an intramolecular Heck cyclization under Pd-catalysis to give tetracyclic 3,4-dihydroquinazolines **52** with four cases in the range of 58–77% yield (Figure 12) [64].

Furthermore, the Ding group explored the DOS of the Ugi/Staudinger/aza-Wittig sequence using a variety of substrates in a stepwise manner. They reported a third example for the synthesis of indolo[1,2-c]quinazolines **56**, achieving 18 examples with yields ranging from 55% to 92% (Figure 13) [75]. The synthesis utilized 2-acylaniline **53** in an Ugi 4-CR. Under the experimental conditions, benzodiazocine **57** was not detected; instead, indolo[1,2-c]quinazoline **56** was obtained. This outcome is likely attributed to the restricted conformation of iminophosphorane **55**, which may be entropically unfavorable for cyclization between the iminophosphorane moiety and the ketone carbonyl group.

Similar to last case for compound **56**, authors developed a DOS example by using benzoylformic acid **58** in Ugi reaction to synthesize multi-functional intermediates **59** with 14 examples ranging from 64% to 91%, then followed by a one-pot Staudinger/aza-Wittig approach to produce 2-acylquinazolines **61** (10 examples, 36–92%) and/or *3H*-1,4-benzodiazepin-3-ones **62** (6 examples, 36–92%) in Figure 14 [76].

Additionally, the Ding group reported a synthesis sequence involving Passerini/Staudinger/aza-Wittig/addition/nucleophilic substitution reactions to produce 3,4-dihydroquinazolines **70**. This sequence begins with the synthesis of azides **64**, yielding 8 examples with yields ranging from 75% to 87% via a three-component Passerini reaction. The azides were then reacted with PPh_3_ and phenyl isocyanates **66** through Staudinger/aza-Wittig reactions to generate carbodiimides **67**. These compounds were subsequently treated with diethylamines **68** to form guanidine intermediates **69**. Finally, under reflux in CH_3_CN with K_2_CO_3_, 18 examples of 3,4-dihydroquinazolines **70** were obtained with yields between 42% and 85%, as illustrated in Figure 15 [77].

Yao and Zhu group reported a novel approach to follow a four-component reaction of Ugi-azide for making intermediates **72** with dual functional sites involving NH and azide groups, then implemented the one-pot synthesis of 3,4-dihydroquinazolines **75** with 17 examples in the range of 56–90% by Pd-catalyzed azide–isocyanide coupling and cyclization (Figure 16) [63].

### 2.3. 2,3-Dihydroquinazolin-4(1H)-One and 4-Quinazolinone

2,3-Dihydroquinazolin-4(1H)-one derivatives have garnered significant attention in pharmaceutical research due to their diverse pharmacological activities. These compounds, often derived from natural sources, serve as pivotal scaffolds in medicinal chemistry, contributing to the development of novel therapeutic agents such as Fenquizone, Quinethazone, Evodiaming, Metolazone, and Febrifugine (Figure 4).

The synthesis of 2,3-dihydroquinazolin-4(1H)-one typically involves the condensation of anthranilic acid derivatives with carbonyl compounds such as aldehydes or ketones. This process often utilizes catalysts, which can include Lewis acids or Bronsted acids, to promote the cyclization and formation of the quinazolinone core [78,79,80]. In another approach, isatoic anhydride is reacted with amines in the presence of a suitable reagent to yield the target compound. The synthesis can be adjusted to incorporate various substituents on the quinazolinone scaffold, allowing for the exploration of its diverse chemical space [81,82]. In addition, the Alves group reported a three-component reaction with 2-azidobenzaldehyde **1a**, phenylacetylene **76a**, and anthranilamide **77a** to make triazoyl-2,3-dihydroquinazolinone **78a** isolated in 82% yield, which was characterized by high- and low-resolution mass spectrometry, ^1^H and ^13^C NMR analysis. This Cu-catalyzed mechanism for making **78a** in dimethyl sulfoxide (DMSO) was proposed: click chemistry with **76a** and azide group of 2-azidobenzaldehyde **1a** was completed to give triazole 82, and followed by an intramolecular cyclization reaction after a nucleophilic attack from the amide nitrogen **82a** to the imine carbon **83** (Figure 17) [83]. They also explored this 3-CR to make heterocycles **85** with 17 examples ranging from 34% to 98% yield, and compounds 87 with 15 examples in the scale of 17–98% through using diamine **84a** and thiourea **86a** instead of **77a**, respectively (Figure 18) [84,85].

Furthermore, similar to 3,4-dihydroquinazolines in biological interests, the versatility of the 4-quinazolinone scaffold allows for varied structural modifications to enhance efficacy and specificity, contributing to its continued interest and exploration in drug discovery (Figure 5). The construction of 4-quinazolinone typically involves cyclization of anthranilic acid or isatoic anhydride [79,81,86]. Based on the Cu-catalyzed synthesis of 3,4-dihydroquinazolines **78a**, the Alves group further offered optimal reaction conditions involving 3-CR of with 2-azidobenzaldehyde **1a**, phenylacetylene **76a**, and anthranilamide **77a** to synthesize 4-quinazolinones **88** with 14 examples ranging from 20% to 67% yield via in situ aromatic oxidation (Figure 19) [83].

### 2.4. 3,4-Dihydroquinazoline-2(1H)-Thione

3,4-Dihydroquinazoline-2(1H)-thione is a heterocyclic compound that has garnered interest in drug discovery due to its potential biological activities. Compounds containing the quinazoline scaffold, including 3,4-dihydroquinazoline derivatives, have been studied for their pharmacological properties, such as anticancer, anti-inflammatory, anti-microbial, anti-malarial, and anti-melanogenesis activities, such as bioactive compounds in Figure 6 [4,87,88,89]. The synthetic route for 3,4-dihydroquinazoline-2(1H)-thione typically involves the cyclization of a precursor, such as anthranilic acid or an isatoic anhydride, with a suitable thiourea component [90,91].

Ding and co-workers developed a Biginelli/Staudinger/aza-Wittig sequence to make 3,4-dihydroquinazolines **44** and **46** (Figure 10) involving a stepwise synthesis of intermediates 42 by Biginelli/Staudinger process. This strategy also facilitates making 3,4-dihydroquinazoline-2(1H)-thiones **90a** and **90b** with 82% and 86% yield through **42** reacting with carbon disulfide (CS_2_) in aza-Wittig reaction, illustrated in Figure 20 [74].

Similar to one-pot sequential Ugi–azide/Pd-catalyzed azide–isocyanide cross-coupling/cyclization reaction to make 3,4-dihydroquinazolines **75** in Figure 15, Ding group also developed a sequential Ugi–azide/Staudinger/aza-Wittig/cyclization reaction to accomplish the stepwise synthesis of 3,4-dihydroquinazoline-2(1H)-thiones **91** with 17 examples ranging from 75% to 94% yield. The whole reaction process underwent Ugi–azide 4-CR to give intermediate **72**, then followed by Staudinger reaction to provide compounds **92**, and aza-Wittig reaction reacting with CS_2_ to obtain adducts **93**, finally afforded compounds **91** via cyclization (Figure 21) [92]. Generally, the synthesis of 3,4-dihydroquinazoline-2(1H)-thiones using CS_2_ suffers from challenges in toxicity, volatility, flammability, handling, and storage difficulties; it is essential to have an insight into human health and environmental hazard [93].

Furthermore, the Zhang group reported two examples of synthesizing 3,4-dihydroquinazoline-2(1H)-thiones using carbon disulfide (CS_2_) through a sequential Staudinger/aza-Wittig/cyclization reaction. The initial example was accomplished via a one-pot process involving a 1,3-dipolar cycloaddition, followed by Staudinger, aza-Wittig, and cyclization reactions. Specifically, intermediate **37** was derived from a three-component [3+2] cycloaddition involving 2-azidobenzaldehyde **1**, amino esters **35**, and maleimides **36** as substrates. This was subsequently followed by Staudinger/aza-Wittig/cyclization reactions without the need for intermediate purification, successfully yielding the 3,4-dihydroquinazoline-2(1H)-thione **94** across 15 examples, with yields ranging from 42% to 73% (Figure 22) [94]. Recently, they developed the second case to make 3,4-dihydroquinazoline-2(1H)-thiones **98** with 16 examples in the scale of 70–93% yield via a one-pot reductive amination/ Staudinger/aza-Wittig/cyclization reaction (Figure 23) [95].

### 2.5. 1,2,3,4-Tetrahydroquinazoline

The core structure of 1,2,3,4-tetrahydroquinazoline and its derivatives is of significant interest due to their diverse biological activities in drug discovery. 1,2,3,4-Tetrahydroquinazoline is a chemical building block that can be synthesized using a variety of methods, typically involving the condensation of an appropriate carbonyl compound with an amine derivative, often in the presence of catalysts. One common approach involves the use of anthranilic acid derivatives and aldehydes, followed by cyclization and reduction steps to produce the tetrahydroquinazoline framework. Zhang group utilized a three-step synthesis to make 1,2,3,4-Tetrahydroquinazolines **103** with 10 examples ranging from 88% to 93% yield, which underwent a one-pot two-step process to make 15 examples of adducts **101** in the scale of 43–73% involving three-component 1,3-dipolar cycloaddition of 2-azidobenzaldehyde **1**, amino esters **35**, and maleimides **36**, [3+2] cycloaddition and denitrogenation of compounds **37** reacting with the second equivalent of maleimides **36′**. Subsequently, 1,2,3,4-Tetrahydroquinazolines **103** were generated by cyclization of adducts **101** reacting with formaldehyde (Figure 24) [96].

## 3. Conclusions

This paper presents 2-azidobenzaldehyde-enabled reactions for synthesizing various quinazoline derivatives, including 3,4-dihydroquinazoline, 2,3-dihydroquinazolin-4(1H)-one, 4-quinazolinone, 3,4-dihydroquinazoline-2(1H)-thione, and 1,2,3,4-tetrahydroquinazoline. These biologically significant quinazoline systems are frequently found in natural products and marketed drugs. The 2-azidobenzaldehyde-initiated approach can be developed into one-pot stepwise syntheses or multicomponent reactions for enhancing operational simplicity, process efficiency, step, pot, and atom economy in sustainability and green chemistry integration. Some of the synthetic methods introduced in this paper provide novel pathways for synthesizing quinazolines, which can also be applied to the synthesis of other heterocyclic blocks. The future of 2-azidobenzaldehyde as a versatile precursor lies in sustainable methodologies, computationally guided design, and the creative exploration of new chemical spaces. Addressing these challenges will enhance its value as a synthetic promoter and drive innovation across multiple scientific disciplines.

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
