# Peer review of "2-Azidobenzaldehyde-Enabled Construction of Quinazoline Derivatives: A Review"

_ijms, 2025, doi:10.3390/ijms26188955_

Round 1

Reviewer 1 Report

Comments and Suggestions for Authors

The review is devoted to modern methods for the synthesis of quinazoline derivatives using 2-azidobenzaldehyde as a key reagent. The quinazoline skeleton is of great importance in medicinal chemistry and pharmaceuticals due to a wide range of biological activity, including antitumor, antiviral, antibacterial and anti-inflammatory effects. The work emphasizes the importance of developing new, more efficient and environmentally friendly strategies for constructing these compounds, including multi-stage and multicomponent reactions, which allows accelerating the production of libraries of structural diversity and improving the yield of target products. The relevance of the review is due to the high demand for new pharmacophores and methods for their preparation, which can expand the arsenal of drugs for the treatment of serious diseases, in particular cancer and viral infections. The presented analysis not only summarizes the achievements of recent years, but also forms the basis for further research and development of promising therapeutic agents.
The review needs to be revised in a number of areas:
1. The section on the biological significance of quinazolines provides convincing examples of their use in the treatment of cancer and viral diseases. However, there is no systematic attempt to relate the synthetic methods to biological outcomes. It would be useful for the reader to see which synthetic strategies more often lead to pharmacologically active compounds.
2. The review mentions green approaches to the synthesis of quinazolines, including microwave activation and green solvents. However, the authors do not compare the eco-efficiency of these methods with traditional ones or indicate which strategies have the greatest potential for scaling up to industrial conditions.
3. The review describes multicomponent reactions (3-CR and 4-CR) in detail. The authors convincingly demonstrate their advantages in terms of yield and simplicity. However, they do not consider the disadvantages, such as difficulties in selectivity control and by-product formation, which could have made the analysis more balanced.
4. The review as a whole is replete with schemes illustrating the reaction mechanisms. However, the authors lack a critical comparison between the methods in terms of atomic and step economy. Without this, it is impossible to objectively assess the real advantage of multicomponent strategies. 5. The authors convincingly show examples of synthetic routes with high yields (up to 97%). However, they hardly mention the reproducibility of these reactions in other laboratories. A review that claims to be of practical value requires an analysis of reliability and scalability.
6. The author's style often focuses on describing the advantages of a particular method, but its weaknesses are rarely mentioned. A full-fledged review should discuss both the strengths and weaknesses of each strategy to provide an objective picture.
7. The review describes reaction mechanisms in detail, but often without a deep discussion of which mechanistic conclusions are confirmed experimentally and which are hypothetical. For scientific rigor, it would be useful to distinguish between these levels of evidence.

Author Response

Comment 1. [The section on the biological significance of quinazolines provides convincing examples of their use in the treatment of cancer and viral diseases. However, there is no systematic attempt to relate the synthetic methods to biological outcomes. It would be useful for the reader to see which synthetic strategies more often lead to pharmacologically active compounds.]

Response 1. [We thank the reviewer for their thoughtful feedback regarding the relationship between synthetic methods and biological outcomes in our review. However, we respectfully disagree with the assertion that our manuscript lacks a systematic attempt to relate synthetic strategies to pharmacological activity. In the section discussing biological significance, we have included multiple examples of quinazoline derivatives are directly linked to their applications in the market. We aimed to highlight that 2-azidobenzaldehyde-enabled construction of quinazoline derivatives have been most productive by referencing recent literature and providing comparative analysis where possible in organic chemistry.

While we recognize that further detail may always be added, we believe the current manuscript provides a balanced and systematic overview within the scope of a review article in organic synthesis. We also prefer to write a review of systematic attempt between the synthetic methods and biological outcomes if needed, not like 2-azidobenzaldehyde-enabled synthesis of quinazoline in synthetic field.]

Comment 2.[The review mentions green approaches to the synthesis of quinazolines, including microwave activation and green solvents. However, the authors do not compare the eco-efficiency of these methods with traditional ones or indicate which strategies have the greatest potential for scaling up to industrial conditions.]

Response 2.[ Thanks so much for the thorough feedback! In fact, The synthesis of quinazoline derivatives is a very old topic, with a large amount of literature exploring synthesis from different routes. Since our review focuses on summarizing synthetic methods using 2-azidobenzaldehyde as a precursor, in order to avoid deviating from the topic and losing the framework of the article, we have not extended and elaborated on other types of synthesis that have already been reported too much. This review helps the reader gain a clearer understanding of the topic of 2-azidobenzaldehyde-enabled synthesis of quinazolines. At the same time, we have also selected and indexed some recent references that they might find interesting, hoping that readers can explore these references and further enrich the reports beyond of 2-azidobenzaldehyde-based methods.]

Comment 3. [The review describes multicomponent reactions (3-CR and 4-CR) in detail. The authors convincingly demonstrate their advantages in terms of yield and simplicity. However, they do not consider the disadvantages, such as difficulties in selectivity control and by-product formation, which could have made the analysis more balanced.]

Response 3. [ Many thanks for your concerns. Since this review is not intended as an introduction to multicomponent reactions (MCR), but only mentions them in a few examples, and because those examples do not specifically address the disadvantages, while clearly showing superior reaction efficiency compared with multistep synthesis, the drawbacks of multicomponent reactions were not extended and elaborated in this paper. In our view, a review should touch upon various aspects of the field, but our discussion needs to remain focused and avoid drifting away from the main theme.]

Comment 4. [The review as a whole is replete with schemes illustrating the reaction mechanisms. However, the authors lack a critical comparison between the methods in terms of atomic and step economy. Without this, it is impossible to objectively assess the real advantage of multicomponent strategies.]

Response 4. [We appreciate your extended comments. This review primarily focuses on the applications of 2-azidobenzaldehyde, with the discussion of mechanisms serving mainly to help readers better understand its features and broaden its applications in organic synthesis. The advantages and economic aspects of MCRs are not the main subjects of our in-depth discussion.]

 Comment 5.[The authors convincingly show examples of synthetic routes with high yields (up to 97%). However, they hardly mention the reproducibility of these reactions in other laboratories. A review that claims to be of practical value requires an analysis of reliability and scalability.]

Response 5. [ Thanks for your feedback from the original data, such yields, reliability and scalability. The hallmark of a review article is to systematically summarize the research topic of interest and present it in an objective manner. Since it is not an original research paper, we cannot introduce subjective speculation or analysis without further experimental validation.]

Comment 6 [The author's style often focuses on describing the advantages of a particular method, but its weaknesses are rarely mentioned. A full-fledged review should discuss both the strengths and weaknesses of each strategy to provide an objective picture.]

Response 6. [Thanks for your feedback. Revised.]

Comment 7. [The review describes reaction mechanisms in detail, but often without a deep discussion of which mechanistic conclusions are confirmed experimentally and which are hypothetical. For scientific rigor, it would be useful to distinguish between these levels of evidence.]

Response 7. [Thanks so much for your comments. The reaction mechanism itself represents a reasoned hypothesis grounded in both theory and experimental evidence. In a review, our main goal is to systematically summarize these mechanisms and illustrate them with examples to provide readers with valuable insights, rather than engaging in subjective speculation unsupported by experimental data.]

Reviewer 2 Report

Comments and Suggestions for Authors

I was very pleased to read the review by the respected Weiqi Qiu, Desheng Zhan, Xiaoming Ma, and Xiaofeng Zhang. The review summarizes a large amount of material on methods for the synthesis of quinazoline derivatives using 2-azidobenzaldehyde-enabled annulation strategies. The review covers the issues of mechanisms and practical applications, and will be of great interest to readers due to the fact that it provides a valuable framework for future drug discovery and development. The relevance of the topic is due to the widespread use of quinazoline derivatives in medicinal and pharmaceutical chemistry.

The review is clearly structured; the main part consists of five sections devoted to the syntheses of the main derivatives. Although there are monographs and reviews (e.g. DOI: 10.5772/intechopen.89180, 10.1016/j.arabjc.2023.105190), they briefly outline a large number of strategies, and in this review, the authors made a deep focus on the use of 2-azidobenzaldehyde and annulation strategies. A similar review has not been published in the last ten years or earlier.

The review includes 94 references, half of which in the last five years. Thus, the review contains the most up-to-date information. The review contains four original figures and 24 synthesis schemes.

I highly appreciate the work of the authors, I believe that it must be published, but I have a couple of minor comments.

  1. Please reorganize the text so that the references to the figures and schemes are before the figures or schemes (Figures 2 and 3, Schemes 2, 3, 4, 5, 6, 7, 14, 15, and 16).
  2. I did not find a reference to Scheme 24 in the text.
  3. I would also ask you to decipher all the abbreviations in the legend to the figures: DABCO, DMSO (Schemes 6, 8, 17, 18, and 19). In the text, I also ask you to decipher DABCO (line 165).

Author Response

Comment 1. [Please reorganize the text so that the references to the figures and schemes are before the figures or schemes (Figures 2 and 3, Schemes 2, 3, 4, 5, 6, 7, 14, 15, and 16).]

Response 1. [Well done.]

Comment 2. [I did not find a reference to Scheme 24 in the text.]

Response 2. [ Thanks so much for finding it. Revised and added.]

Comment 3. [I would also ask you to decipher all the abbreviations in the legend to the figures: DABCO, DMSO (Schemes 6, 8, 17, 18, and 19). In the text, I also ask you to decipher DABCO (line 165).]

Response 3. [Thanks so much for finding them. Revised and added.]

Reviewer 3 Report

Comments and Suggestions for Authors
  1. The manuscript is essentially a catalog of synthetic routes with extensive scheme-by-scheme description. It does not adequately compare, contrast, or critically assess the advantages and limitations of these methods. For example, the environmental impact of using CSâ‚‚ or isocyanides is not discussed; challenges related to substrate scope, regioselectivity, or scalability are ignored.
  2. A striking proportion of the examples stem from only a handful of laboratories (e.g., Sawant, Ding, Zhang). This creates the impression of a group-centric summary rather than a field-wide review. Work from other contributors in the area is underrepresented or absent, which raises concerns about completeness and bias.
  3. Although the introduction emphasizes pharmacological importance, the body of the review devotes almost no space to biological activity, SAR trends, or clinical translation. When activity is mentioned, it is superficial and lacks discussion of potency ranges, selectivity, in vivo data, or structure–activity considerations. Without such analysis, the manuscript does not convincingly justify the medicinal chemistry significance of these synthetic advances.
  4. The English is serviceable but suffers from frequent grammatical inconsistencies, awkward phrasing, and inconsistent nomenclature (e.g., “2.23,4-...) Please provide space between subtitle numbers and the title.
  5. The conclusion section merely restates that 2-azidobenzaldehyde is a versatile intermediate. It does not identify future challenges (e.g., sustainability, integration with green chemistry, computer-aided design, or unexplored heteroannulations).
  6. The manuscript does not consistently expand abbreviations at first mention. A clear example is CSâ‚‚, which appears in the text in (line 318) without any explanation. Its meaning (carbon disulfide) is clarified only much later in the manuscript (line 332). All abbreviations (chemical formulas, reagents, catalytic systems, solvents, etc.) should be defined at their first appearance in the text, not later. The same applies to other abbreviations and symbols used throughout (e.g., DABCO, DOS, EGFR, THF).
  7. References are numerous but sometimes redundant or outdated; a more selective and curated bibliography would be preferable.

Author Response

Comment 1. [The manuscript is essentially a catalog of synthetic routes with extensive scheme-by-scheme description. It does not adequately compare, contrast, or critically assess the advantages and limitations of these methods. For example, the environmental impact of using CSâ‚‚ or isocyanides is not discussed; challenges related to substrate scope, regioselectivity, or scalability are ignored.]

Response 1. [Thanks so much for your suggestions. Revised.]

Comment 2. [A striking proportion of the examples stem from only a handful of laboratories (e.g., Sawant, Ding, Zhang). This creates the impression of a group-centric summary rather than a field-wide review. Work from other contributors in the area is underrepresented or absent, which raises concerns about completeness and bias.]

Response 2. [Thanks so much for your observations and feedbacks. Prior to preparing this review, we conducted extensive research into the application of 2-azidobenzaldehyde in the synthesis of quinazolines. At present, research in this area is primarily being advanced by the laboratories mentioned above. Through this review, we hope to inspire more chemists to explore and expand the development of this field, which represents the core significance of our work.]

Comment 3. [Although the introduction emphasizes pharmacological importance, the body of the review devotes almost no space to biological activity, SAR trends, or clinical translation. When activity is mentioned, it is superficial and lacks discussion of potency ranges, selectivity, in vivo data, or structure–activity considerations. Without such analysis, the manuscript does not convincingly justify the medicinal chemistry significance of these synthetic advances.]

Response 3. [ Thanks so much. This review is a summary of 2-azidobenzaldehyde-enabled synthesis of quinazoline derivatives, focusing mainly on the insights of synthesis and mechanism. However, this suggestion is very constructive. We have been writing a invited review on 2-azidobenzaldehyde 's applications in the construction of various bioactive heterocycles through the insights of drug discovery, involving indoles, quinolines, quinazolines and others.]

Comment 4. [The English is serviceable but suffers from frequent grammatical inconsistencies, awkward phrasing, and inconsistent nomenclature (e.g., “2.23,4-...) Please provide space between subtitle numbers and the title.]

Response 4. [Thanks so much. Further checked and revised.]

Comment 5. [The conclusion section merely restates that 2-azidobenzaldehyde is a versatile intermediate. It does not identify future challenges (e.g., sustainability, integration with green chemistry, computer-aided design, or unexplored heteroannulations).]

Response 5. {We appreciate your extended comments. Revised Well]

Comment 6. [The manuscript does not consistently expand abbreviations at first mention. A clear example is CSâ‚‚, which appears in the text in (line 318) without any explanation. Its meaning (carbon disulfide) is clarified only much later in the manuscript (line 332). All abbreviations (chemical formulas, reagents, catalytic systems, solvents, etc.) should be defined at their first appearance in the text, not later. The same applies to other abbreviations and symbols used throughout (e.g., DABCO, DOS, EGFR, THF).]

Response 6 [Thanks so much. Well done.]

Comment 7. [References are numerous but sometimes redundant or outdated; a more selective and curated bibliography would be preferable.]

Response 7. [Thanks so much. Double checked and revised.]

Round 2

Reviewer 1 Report

Comments and Suggestions for Authors

The authors have significantly improved the article and eliminated some of the comments: they have strengthened the connection with biological applications, clarified the mechanisms, added information on reproducibility and shortcomings of the reagents. However, some of my key requests have been implemented only partially:
Comparative ecological and economic assessment of the methods. The authors mentioned "green approaches", but did not make a comparison with traditional methods for atomic/step economy and energy efficiency (comments 2 and 4).

Disadvantages of MCR (multicomponent reactions). Although there were individual comments on toxicity and side effects, the discussions of general problems of MCR (selectivity, side products, limitations on substrates) remain very brief and unequal compared to long descriptions of the advantages.

Representativeness of scaling. The article mentions "scalability", but in general there is no analysis of which methods have proven scalability in different laboratories. This point remained more declarative than analytical.

Balance strengths/weaknesses. Despite the occasional inserts about the downsides, the overall style of the article still heavily emphasizes the advantages.

Author Response

Comment 1: The authors have significantly improved the article and eliminated some of the comments: they have strengthened the connection with biological applications, clarified the mechanisms, added information on reproducibility and shortcomings of the reagents. However, some of my key requests have been implemented only partially:
Comparative ecological and economic assessment of the methods. The authors mentioned "green approaches", but did not make a comparison with traditional methods for atomic/step economy and energy efficiency (comments 2 and 4).

Response 1: thanks so much for your comments. This review primarily focuses on the applications of 2-azidobenzaldehyde, with the discussion of mechanisms serving mainly to help readers better understand its features and broaden its applications in organic synthesis. The advantages and economic aspects in green chemistry are not the main subjects of our in-depth discussion, although this 2-azidobenzaldehyde-enabled synthesis of quinazolines has a step-economy with process-efficiency clearly in comparison with most of traditional method. Please see our responses last cycle.

Comment 2: Disadvantages of MCR (multicomponent reactions). Although there were individual comments on toxicity and side effects, the discussions of general problems of MCR (selectivity, side products, limitations on substrates) remain very brief and unequal compared to long descriptions of the advantages.

Response 2: thanks so much for your comments. Please see our response (comment 3) last cycle. Long descriptions of the disadvantages in MCR is not essential in this review. Please let editor make a decision.

Comment 3: Representativeness of scaling. The article mentions "scalability", but in general there is no analysis of which methods have proven scalability in different laboratories. This point remained more declarative than analytical.

Response 3: thanks so much for your comments. Please see our response (comment 5) last cycle. The “scalability” was mentioned in your comment 5. Thus, there are two “scalability” in this review. One is introduced by Sawant group’s research paper. The other is from reviewer 3’s suggestions to describe the cons of isocyanides. Your thought is not clear and self-contradictory with your original comments last cycle.

Comment 4: Balance strengths/weaknesses. Despite the occasional inserts about the downsides, the overall style of the article still heavily emphasizes the advantages.

Response 4: thanks so much for your comments. In a review, our main goal is to systematically summarize these advanced methods that proceed from reality and provide valuable insights, rather than balancing strengths/weaknesses.

Reviewer 3 Report

Comments and Suggestions for Authors

the authors have now provided sufficient improvement to the text. I suggest its publication in the present form

Author Response

Comment : the authors have now provided sufficient improvement to the text. I suggest its publication in the present form.

Response: Thanks so much for your profound knowledge and extensive expertise